

# Twisted tale of the tiger: the case of inappropriate data and deficient science

Qamar Qureshi[1],[*], Rajesh Gopal[2] and Yadvendradev Jhala[1],[*]

[1] Wildlife Institute of India, Dehradun, Uttarakhand, India
[2] Global Tiger Forum, New Delhi, Delhi, India
[*] These authors contributed equally to this work.

## ABSTRACT

Publications in peer-reviewed journals are often looked upon as tenets on which future scientific thought is built. Published information is not always flawless and errors in published research should be expediently reported, preferably by a peer-review process. We review a recent publication by Gopalaswamy et al. (10.1111/2041-210X.12351) that challenges the use of "double sampling" in large-scale animal surveys. Double sampling is often resorted to as an established economical and practical approach for large-scale surveys since it calibrates abundance indices against absolute abundance, thereby potentially addressing the statistical shortfalls of indices. Empirical data used by Gopalaswamy et al. (10.1111/2041-210X.12351) to test their theoretical model, relate to tiger sign and tiger abundance referred to as an Index-Calibration experiment (IC-Karanth). These data on tiger abundance and signs should be paired in time and space to qualify as a calibration experiment for double sampling, but original data of IC-Karanth show lags of (up to) several years. Further, data points used in the paper do not match the original sources. We show that by use of inappropriate and incorrect data collected through a faulty experimental design, poor parameterization of their theoretical model, and selectively picked estimates from literature on detection probability, the inferences of this paper are highly questionable. We highlight how the results of Gopalaswamy et al. were further distorted in popular media. If left unaddressed, the paper of Gopalaswamy et al. could have serious implications on statistical design of large-scale animal surveys by propagating unreliable inferences.

## INTRODUCTION

The scientific method operates by testing competing hypothesis or by choosing between alternate models that best explain observed data. Hypothesis and models that survive repeated testing by careful experimentation are published through rigorous scrutiny by a peer-review process, these subsequently become scientific theory (*Gauch, 2012*).

An incorrect experimental design, inappropriate data collection protocol, and selective data used for analysis from telemetered Florida panthers (*Puma concolor*) (*Gross, 2005*) resulted in a peer-reviewed publication of habitat use and preference (*Maehr & Cox, 1995*) in Conservation Biology. The results were subsequently used for land use planning

Corresponding author
Yadvendradev Jhala,
yvjhala@gmail.com

and policy (*Maehr & Deason, 2002*) which resulted in the best panther habitat being lost to developmental projects (*Gross, 2005*). In an ideal world, response to deficiencies in science is best made through a peer-review process, since scientists understand the intricacies of the scientific method probably more than others (*Parsons & Wright, 2015*).

In a recent paper "An examination of index-calibration experiments: counting tigers at macroecological scales" published in the journal *Methods of Ecology and Evolution*, *Gopalaswamy et al. (2015a)* supposedly demonstrate that as part of their long-term, large-scale data on tiger abundance and index (IC-Karanth) they did not find any relationship between tiger abundance and scat index. They conclude that attempting to use double sampling (*Cochran, 1977*; *Eberhardt & Simmons, 1987*; *Pollock et al., 2002*) to establish relationships between any index of abundance and actual abundance is a futile effort. In particular, they claim that the relationship between tiger sign index and tiger abundance published by *Jhala, Qureshi & Gopal (2011a)* to be improbable since they could not reproduce it by their data or theoretical model. We review *Gopalaswamy et al. (2015a)* to show that by the use of (a) wrong ecological parameters for their theoretical model, (b) selectively picked references from literature, (c) inappropriate and incorrect data, and (d) data not collected in an experimental setup, the inferences drawn by their paper are questionable.

## USE OF INADEQUATE ECOLOGICAL PARAMETERS

The basic premise for index calibration by double sampling is that animal sign intensity or count data should reflect underlying animal abundance. Often due to logistic and economic constraints large-scale estimates of abundance are not possible through statistically rigorous methods that explicitly estimate and correct for detection (e.g. capture-mark-recapture or DISTANCE sampling). Double sampling approach as described initially by *Cochran (1977)* and applied to wildlife surveys by *Eberhardt & Simmons (1987)*, allows us to address this limitation by measuring a relatively easy and economically less expensive, but potentially biased index of abundance across all sampling units, while simultaneously estimating detection corrected abundance from within a subset of these sampling units (*Conroy & Carroll, 2009*; *Williams, Nichols & Conroy, 2002*). Subsequently, the potentially biased index is calibrated against the unbiased estimate of abundance or actual abundance using a ratio or regression approach (*Skalski, Ryding & Millspaugh, 2005*). *Pollock et al. (2002)* recommend double sampling as a sensible large-scale survey design for most species.

To prove their point of view, *Gopalaswamy et al. (2015a)* use detection probability ($p$) estimates from tiger occupancy studies as a surrogate for detection probability of tiger scat for parameterizing their theoretical model. This $p$ is the probability of finding (or not finding) tiger sign on a single survey in an area occupied by tigers. *Gopalaswamy et al. (2015a)* confuse $p$ of occupancy surveys with the probability of finding (or missing) an individual sign (in this case tiger scat) (r). In other words, $p$ represents the number of surveys out of the total surveys (proportion) that are likely to detect the presence of tigers in an occupied site, while $r$ represents the proportion of tiger signs that are detected (or missed) in a single survey. The two are not the same

i.e. $p \neq r$. For example, a survey that detected nine out of 10 signs present or another that detected one sign out of 10 signs are both considered as having 100% detection of tiger presence ($p = 1$) for an occupancy survey, but $r$ for each of these surveys is 0.9 and 0.1 respectively. Thus, detection probability ($p$) of occupancy surveys is not informative on per capita detection rates ($r$) of tiger sign. For estimating $r$ the correct approach would be to use a double blind observer experimental design (*Buckland, Laake & Borchers, 2010*; *Nichols et al., 2000*), where two observers would walk the same trail some distance apart and record observed tiger scat without communicating with each other. The scats being missed by each of them could then be used to estimate the probability of missing scats entirely.

Also, in occupancy surveys all kinds of signs are often used to detect tigers (pugmarks, scat, scrape, rake marks, direct sightings, vocalization, tiger kills, etc). *Karanth et al. (2011a)* have used both tiger scat and tiger pugmark to detect tigers in a grid for estimating occupancy. Thus, detection probability of occupancy in these surveys is the compounded probability of occurrence and detection of both scat and pugmark on a single survey which cannot be teased apart and used as a surrogate for detecting individual scats. From the above it is clear that the use of occupancy detection probability to parameterize detection probability of tiger scat in the theoretical model of *Gopalaswamy et al. (2015a)* is wrong. Typically in a double sample survey the index is measured without an estimate of its detection, by calibrating this potentially biased index against abundance, double sampling elegantly addresses the issue of detection and other sources of variability in the index (*Conroy & Carroll, 2009*).

## SELECTIVELY PICKED REFERENCES

Not only do *Gopalaswamy et al. (2015a)* use an incorrect detection probability (derived for occupancy studies) in place of a double observer-based detection probability for sign intensity for their theoretical model, they were selective in picking low estimates of detection probability with high coefficient of variation (CV) from those available in published literature. The estimates of detection probability $p$ at one km segments (0.17) and its CV (1) from *Karanth et al. (2011a)* were used, claiming that these were the only parameter estimates available. The use of low $p$ and extraordinarily high CV to suggest that detection of tiger presence for occupancy survey is in general low and highly variable. These parameters play an important role in subsequent derivations in the paper. *Gopalaswamy et al. (2015a)* have ignored other published estimates of these parameters obtained by sampling large areas and derived by following the same field and analytical protocols. These publications report far higher $p$ with much smaller CV (*Harihar & Pandav (2012)*, $p = 0.951$ SE 0.05; *Barber-Meyer et al. (2013)*, $p = 0.65$ SE 0.08). The low $p$ and high CV reported by *Karanth et al. (2011a)* is likely due to poor design and not a norm in detecting tiger presence. In our experience tigers uses scat, scrape, and rakes to advertise their presence and it is highly unlikely that tiger signs will have such a low detectability unless the population is very low, survey design is poor, or data are collected by inexperienced/untrained persons.

## INAPPROPRIATE AND INCORRECT DATA

Throughout the paper the authors have used data and parameters related to tigers published by K. Ullas Karanth (a co-author on the paper) and colleagues, which they refer to as Index-calibration experiment—(IC-Karanth). The authors have presented eight paired data points on tiger density and tiger signs (in fact only scats) in figure 5 of the paper. This graph shows no relationship between tiger scat encounter rate and tiger density, considered as an empirical test in support of their theoretical model based only on eight data points. On perusal of the references cited in *Gopalaswamy et al. (2015a)*, we noticed several irregularities which invalidate the use of these data as a scientific experiment to test this relationship. It is relevant to point out that for calibration of any index with abundance as done in a double sampling experimental approach (*Eberhardt & Simmons, 1987*), both index and abundance, should be sampled contemporaneously and over the same spatial extent (paired in time and space). In three data points out of eight presented in figure 5 of *Gopalaswamy et al. (2015a)*, tiger signs and tiger density were not collected contemporaneously. Tiger density can fluctuate substantially between years (*Karanth et al., 2006*) and tiger signs have short persistence time. Yet, the data *Gopalaswamy et al. (2015a)* use for their paired experiment has lags of several years (2–7 years) between estimating tiger density and tiger sign (Fig. 1). In particular, the data point from Bandipur has a lag of 7 years (density estimated in 1999, scat sampling in 2006), data point representing Melghat has a lag of 3 years (density estimated in 2002, scat sampling done in 2005) and data point from Pench Maharashtra has a lag of 2 years (density estimated in 2002, scat sampling done in 2004) (*Karanth & Nichols, 2000*, *2002*; *Karanth et al., 2004*; *Karanth & Kumar, 2005*; *Andheria, 2006*, see Supplemental Material for relevant sections of these publications). The authors do have concurrent density estimates from one of these sites (Bandipur) with smaller variance (*Gopalaswamy et al., 2012*), but curiously have not chosen to use or refer to this. At one data point (Tadoba), an extreme outlier at right corner of figure 5 of *Gopalaswamy et al. (2015a)* (Fig. 1), the data on scat encounters does not match the original source (scat encounter rate 3.6/10 km as given in figure 5 of *Gopalaswamy et al. (2015a)* vs. 1.99/10 km as given in the original source (*Karanth & Kumar, 2005*; but addressed this by mentioning that the original reference was incorrect in a corrigendum to the original paper *Gopalaswamy et al. (2015b)*). Yet, two data points (Melghat and Pench Maharashtra) continue to differ in their Fig 5 (*Gopalaswamy et al., 2015a)* from the cited references in the corrigendum *Gopalaswamy et al. (2015b)*.

Methods for recording scat encounter rates differed between source reference sites used for IC-Karanth. *Andheria (2006)* removed all scats encountered on the first sample and discarded them from data analysis, a practice which is not uniformly followed for recording tiger scat encounter rates in other studies. For studies referenced for IC-Karanth, camera-trap sampling was done in small areas within larger protected areas for estimating tiger density, whereas tiger scats were collected for studying tiger diet (*Karanth & Nichols, 2000*, *2002*; *Karanth et al., 2004*; *Karanth & Kumar, 2005*; *Andheria, 2006*) possibly opportunistically from the entire reserve. Any intent of calibrating these tiger scat
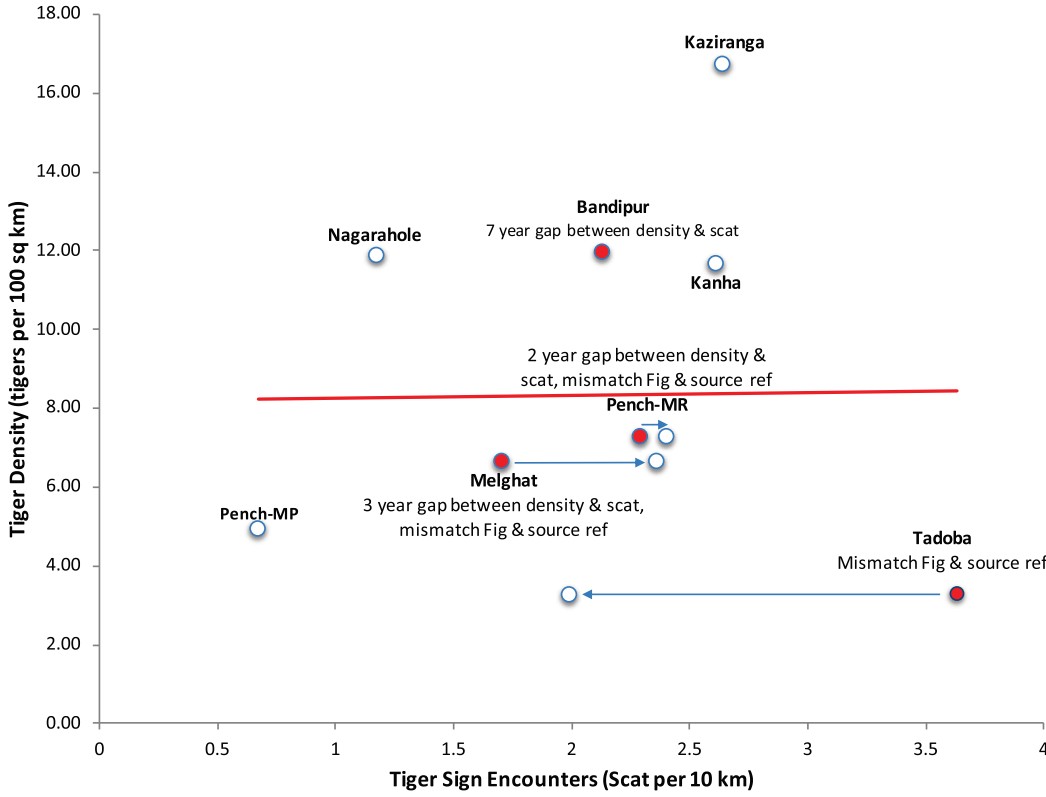

**Figure 1** Recreation of Figure 5 from *Gopalaswamy et al. (2015a)* **highlighting the data discrepancies in the index-calibration experiment.** The names of tiger reserves from central Indian landscape and Western Ghat landscape, where sampling was done are mentioned. MR refers to the State of Maharashtra, and MP refers to the State of Madhya Pradesh.

data to tiger density obtained through camera-trap sampling is not mentioned in any of the original sources. In the original studies cited by *Gopalaswamy et al. (2015a)* referred to as IC-Karanth experiment, there seems to be no intent of designing an experiment to evaluate the relationship between tiger sign encounter rate and tiger density, the sources are unclear if the scat sampling was done within the same spatial extent as the camera-trap survey for estimating tiger density. The basic premise of a double sampling experimental approach, wherein data from both samples (index and density) need to be paired in time and space is violated in the field experiment (IC-Karanth) of *Gopalaswamy et al. (2015a)* invalidating their conclusions.

## VARIABILITY IN TIGER CAPTURE PROBABILITY AND DENSITY ESTIMATES FROM CAMERA-TRAP CAPTURE-MARK-RECAPTURE

As with occupancy detection probability, *Gopalaswamy et al. (2015a)* restrict themselves entirely to 11 estimates of tiger density published by *Karanth et al. (2004)* for their models. On multiple occasions they point out the highly variable capture probability $p$ and variance associated with tiger density estimates. In fact, in light of the large number of published tiger density estimates with higher precision (e.g. 21 estimates in

*Jhala, Qureshi & Gopal, 2011a*), these authors should have considered *Karanth et al. (2004)* estimates as particularly lacking in precision. When, estimates with large sampling errors are used to guide development of theoretical models it would be difficult to deduce any relationship between tiger signs and tiger density. Poor precision of tiger density estimates in *Karanth et al. (2004)* were likely due to poor sampling design and not something that is inherent in tiger population estimation, e.g. for data presented in *Karanth et al. (2004)* CV of tiger density increases with increase in sampled area and $p$ decreases with the sampled area ($r = 0.4$ and $-0.63$ respectively). Overstating the case of sampling uncertainty can only do harm to the development and adoption of sound and practical methods.

## REPEATING NON PEER-REVIEWED LITERATURE TO ADVANCE UNSUBSTANTIATED CLAIMS

*Gopalaswamy et al. (2015a)* claim that the methods followed by *Jhala, Qureshi & Gopal (2011a)* have resulted in "improbable estimates of 49% increase in tiger density over 4 years". *Gopalaswamy et al. (2015a)* do not explain how they arrived at the figure of 49% increase, they cite a letter to Science, commenting on a news article (*Karanth et al., 2011b*), but they have not explained the 49% increase in tiger abundance in this letter as well (*Jhala, Qureshi & Gopal, 2011c*). The fact is that in 2006 India's mean tiger population was estimated at about 1,400 while in 2010 the estimate was about 1,700 but included estimates from some new areas like Sundarbans that were not assessed in 2006. Comparing tiger numbers between common areas sampled in 2006 and 2010 an increase of 17.6% was estimated in 4 years, or about 4% per year; which is very probable for large carnivores. It is inexplicable to us how *Gopalaswamy et al. (2015a)* arrived at a 49% increase in abundance or why they continue to perpetuate this obviously erroneous inference.

## PROPAGANDA THAT IS NOT CONSISTENT WITH FACTS

The paper of *Gopalaswamy et al. (2015a)* is, as the title suggests, about "index calibration experiment" especially referring to estimation of tiger abundance. To this extent the reference to *Jhala, Qureshi & Gopal (2011a)* that demonstrates a strong relation between tiger sign index and tiger abundance as IC-Jhala and several publications of U. Karanth as IC-Karanth is relevant. *Gopalaswamy et al. (2015a)* seem to have gone through the methods employed for estimating the status of tigers in India thoroughly (*Jhala, Qureshi & Gopal, 2008*, *2011b*, *2015*), since they have meticulously computed parameters from these reports for their paper. K. U. Karanth is also an author on several chapters in *Jhala, Qureshi & Gopal (2015)*. They should know that national tiger status assessments (*Jhala, Qureshi & Gopal, 2008*, *2011b*, *2015*) were never based on tiger sign index alone. Tiger sign index was one amongst the many ecologically important covariates that included human footprint, prey abundance, and landscape characteristics that were used for modeling tiger density. Yet, the blog of the journal Methods in Ecology and Evolution titled "flawed method puts tiger rise in doubt" states "amongst recent studies thought to be based on this method is India's national tiger survey" (*Grives, 2015*) which the blog then discredits as being inaccurate based on conclusions of *Gopalaswamy et al.*

*(2015a)*. The fact is India's national tiger survey of 2014 (*Jhala, Qureshi & Gopal, 2015*) used spatially explicit capture-recapture (SECR) in a joint likelihood-based framework (*Efford, 2011*) with covariates of prey abundance, tiger sign intensity, habitat characteristics, and human footprint. The SECR and Joint likelihood analysis are a recent development (*Borchers & Efford, 2008*; *Efford, 2011*) and therefore could not have been used for earlier national tiger assessments which used general linear models (*Jhala, Qureshi & Gopal, 2008*, *2011b*).

The misleading reports that subsequently followed in the media had forgotten that the MEE paper by *Gopalaswamy et al. (2015a)* is a debate on index calibration using double sampling approach (*Eberhardt & Simmons, 1987*) with simple linear regression and not about national tiger status assessment. The 2014 national tiger status assessment was based on photo-captures of 1,506 individual tigers, capture-histories of these were subsequently modeled in SECR with covariates of prey, habitat, and human impacts to estimate 2,226 (SE range 1,945–2,491, >1.5 year old) tigers from across India (*Jhala, Qureshi & Gopal, 2015*). This amounts to 68% of the total tiger population being photo-captured and 77% (1,722; 95% CI [1,573–2,221] tigers) of the total tiger population being estimated by capture-mark-recapture without any extrapolation using covariates/indices. By muddling index calibration with the national tiger survey in the paper (*Gopalaswamy et al., 2015a*) and in all subsequent press releases and interviews Dr. Ullas Karanth and coauthors incorrectly use the *Gopalaswamy et al. (2015a)* paper results (which are themselves highly questionable) to discredit the national tiger survey results as being inaccurate (*Bagla, 2016*; *Chauhan, 2015*; *Croke, 2015*; *Grives, 2015*; *Karanth, 2015*, *2016*; *Rohit, 2015*; *Sinha & Bhattacharyal, 2015*; *Varma, 2015*; *Vaughan, 2015*; *Vishnoi, 2015*) and mislead the readers.

Peer-reviewed publications form the basis for advancement of science and are often cited and used as a basis from which to move ahead. Indeed, the *Gopalaswamy et al. (2015a*, *2015b)* paper has been subsequently cited in papers addressing methodological reviews, advances and policies (*Darimont et al., 2018*; *Hayward et al., 2015*), abundance estimation papers (*Broekhuis & Gopalaswamy, 2016*; *Caley, 2015*; *Elliot & Gopalaswamy, 2017*; *Falcy, McCormick & Miller, 2016*; *Mahard et al., 2016*) and in some Masters and PhD thesis (*Walker, 2016*; *Moorcroft, 2017*). Published scientific literature can have errors, these can occur through negligence of scientists or deliberate misleading of science (*Macilwain, 2014*), and can pass the peer-review process due to ignorance, poor diligence, or vested interest (*Parsons & Wright, 2015*). Mistakes in published science should be corrected expediently, as these are detrimental to the scientific progress in the specific field and propagate a wrong basis for further research. In our opinion, *Gopalaswamy et al. (2015a*, *2015b)* results are misleading, due to inappropriate scientific process and data, and have therefore not contributed to the wider debate on the usefulness of double sampling (*Eberhardt & Simmons, 1987*; *Pollock et al., 2002*) for large-scale animal surveys.

We stress that landscape scale surveys need to be a blend of robust statistical design and analysis that are pragmatic (economic and logistically possible) to achieve. The national tiger surveys of India (*Jhala, Qureshi & Gopal, 2008*, *2011b*, *2015*) have striven to keep pace with modern advancement in animal abundance techniques and analysis and

have used robust statistical tools available within the constraints of large-scale data coverage, resources, and timeframe. The concept and philosophy of double sampling (*Cochran, 1977*) form the basis for modern statistical and analytical approaches that infer relationships between actual abundance and counts, indices, and covariates. The family of general linear models, generalized additive models (*Zuur et al., 2009*), joint likelihood (*Conroy et al., 2008*), SECR with habitat covariates (*Efford & Fewster, 2013*), and SECR joint likelihood (*Chandler & Clark, 2014*) take the relationship between an index/covariates and absolute abundance to various levels of analytical complexity. There seems to be some agreement on the best analytical approach to use for landscape scale abundance estimation of tigers between *Gopalaswamy et al. (2015a)* and us (*Jhala, Qureshi & Gopal, 2015*). *Gopalaswamy et al. (2015a)* recommend using the joint likelihood approach, while the tiger status assessment for India for the year 2014 used spatially explicit joint likelihood with camera-trap data of tigers, and covariates of tiger sign index, prey abundance, and human footprint indices (*Jhala, Qureshi & Gopal, 2015*). Yet, we stress the relevance and importance of first exploring relationships of abundance with indices and covariates, based on sound ecological logic before attempting complex statistical analysis, and refrain from putting the proverbial cart (statistical) before the horse (ecology) (*Krebs, 1989*).

## ACKNOWLEDGEMENTS

We acknowledge Rashid Raza for painstakingly retrieving old reports to verify data used for IC-Karanth and helping draft this paper. We thank S. Dutta, S. Bist, Bipin C., and V. Kolipakam for their comments on the manuscript.

### Funding

The authors received no funding for this work.

### Competing Interests

Yadvendradev Jhala and Qamar Qureshi are employed by the Wildlife Institute of India and Rajesh Gopal is employed by the Global Tiger Forum.

### Author Contributions

- Qamar Qureshi conceived and designed the experiments, performed the experiments, analyzed the data, contributed reagents/materials/analysis tools, prepared figures and/or tables, authored or reviewed drafts of the paper, approved the final draft.
- Rajesh Gopal contributed reagents/materials/analysis tools, authored or reviewed drafts of the paper, approved the final draft.
- Yadvendradev Jhala conceived and designed the experiments, performed the experiments, analyzed the data, contributed reagents/materials/analysis tools, prepared figures and/or tables, authored or reviewed drafts of the paper, approved the final draft.

## Data Availability

All relevant information is in the article and the Supplemental File. The research in this article did not generate any data or code as it is a critique of a published paper.

## Supplemental Information

Supplemental information for this article can be found online at http://dx.doi.org/10.7717/peerj.7482#supplemental-information.

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
