# Peer review of "Twisted tale of the tiger: the case of inappropriate data and deficient science"

_PeerJ, doi:10.7717/peerj.7482_

## Round 0.1 · original submission · Major Revisions

Your MS has received three detailed reviews from independent experts, as well as an assessment from A. Gopalaswamy (the author of the study you critiqued). The three independent (anonymous) reviews were thorough and thoughtful. They provide advice to the authors on how to recast the reply to make the language less confrontational or informal, as well as helpful advice from the all referees about how to better present the critical arguments and re-analysis, were valuable. Gopalaswamy explains some of the backstory regarding the back-and-forth critiques and the editorial impasse created in Methods in Ecology and Evolution.

Given the history of this debate, and the fact that the paper being critiqued did not appear originally in PeerJ, this presents somewhat of an editorial conundrum. However, the presentation of scientific debate on India’s most iconic species would undoubtedly serve as a valuable resource to the broader ecological community on approaches to and scientific challenges facing conservation management within vexed contexts. With this consideration in mind, we would like to invite you to submit a revised MS for consideration, which recasts it as a research article (this is important to bring it 'in scope' for PeerJ), tones down the potentially inflammatory rhetoric, and takes careful account of the independent referee suggestions, as well as respectfully addresses the criticisms in the rejoinder of Gopalaswamy.

Reviewer 1 ·

Basic reporting

I few references provided in the text need to be clarified:
Line 65: “…Jhala et al. (2011)…” is it a or b? Clarify please.
Lines 119-120: “…Karanth et al. (2011)…” which one a or b? Clarify please.
Lines 127-128: “…Karanth et al. (2011)…” which one a or b? Clarify please.

Some important information is missing in the supplementary material. This would also ease the work of the reviewers. Please add the information regarding abundance estimates by means of photographic capture-recapture for Pench-MP, Nagarahole, Kaziranga, Kanha and Tadoba in the suppl. material. Same comment for the encounter rate for Pench-MP, Nagarahole, Bandipur (the value provided in Fig. 1 of the present manuscript does not correspond to the value provided in the suppl. material; see also my comment Figure 1; clarify please), Kaziranga and Kanha.

Experimental design

Although "experiments" in Bandipur, Melghat and Pench-MR have lags of several years between estimating tiger densities and tiger signs and hence should not be included in the experiment/graph (e.g. Figure 1), they do furthermore not match the original source. Clarify please and correct where necessary..

Validity of the findings

The manuscript does not answer if double sampling to establish relationship between any index of abundance and actual abundance is a worthwhile effort. The argumentation would gain in strength if the theoretical model e.g. developed by Gopalaswamy et al. (2015) would be parameterized using detection probability of tiger signs instead of detection probabilities to predict the R2 statistic based on the theoretical models and if data that demonstrate that there is in fact a strong positive relationship between tiger abundance and scat index would be presented. I encourage to include these aspects in the present manuscript.

Additional comments

Major comments
Lines 125-127: The detection probability varies with the size of the sample unit (e.g. 225 km2 in Barber-Meyer et al. 2012; 166 km2 in Harihar and Pandav 2012; 188 km2 in Karanth et al. 2011a) and the definition of the sampling occasion (e.g. Barber-Meyer et al. maximum of 40 km was surveyed per grid cell with each contiguous 1-km segment considered a ‘spatial replicate’; according to Harihar and Pandav maximum survey distance as 40 km, the data were recorded along segments of 250 m as either detected or not-detected, detection histories were constructed for each cell by aggregating signs along 250 m segments at 1 km to form ‘replicates; Karanth et al. 2011b fixed the total sampling effort (distance walked) in a cell with 100%habitat at 40 km, each type of sign detection was assigned only once to each 100-m trail segment, thus yielding the standard ‘1’ (detection) or ‘0’ (nondetection) histories required for occupancy analyses. These sign detection data were aggregated at 1 km length to form ‘spatial replicates’) considered in a given survey/study. Hence when attempting to compare detection probability estimates across multiple studies it is usually necessary to make sure that the per occasion sampling effort is standardised across sampling areas. Please make sure that the detection probabilities are comparable. If not please adjust them according to the procedure described e.g. in Gopalaswamy et al. (2015).
Line 150: Scats were sampled in 2006 in Bandipur according to the information provided in the supplementary material. Correct please.
Lines 177-181: It does not make sense to calculate a coefficient of determination with only 4 points (e.g. Figure 1 in the manuscript). Ideally there should be at least 10 points. More over only three data points do not qualify and there is no reason not to include the value provided in the original source (namely 1.99/10 km) of the extreme outlier at the right corner of figure 5 in the calculation of the coefficient of determination! If this value is included the coefficient of determination will be lower (athough the sample size is still too low). Although "experiment" in Bandipur, Melghat and Pench-MR have lags of several years between estimating tiger densities and tiger signs and hence should not be included in the experiment/graph, they do furthermore not match the original source. Clarify and adjust the values where necessary (see my comment Figure 1).
The argumentation would gain in strength if it would be possible to demonstrate that there is in fact a strong positive relationship between tiger abundance and scat index. Ideally data (abundance and index collected contemporaneously and over the same spatial extent) collected according to the same design and in the same landscapes (as it is possible that scat detectability varies from one landscape to the other) showing that there is in fact a strong positive relationship between tiger sign index and tiger abundance should be presented (see also my comment "validity of the finding").

Figure 1: I see some additional discrepancies in the values presented in Figure 1. According to the information presented in the suppl. material the encounter rates (scats per 10 km) are 2.36 in Melghat (year 2005), 2.12 in Bandipur (mean encounter rate of scats; year 2006) and 2.4 in Penchar-MR (year 2004), which differ from the values presented in Figure 1. Clarify please and adjust where necessary.

Minor comments
Line 67: a space is missing before “cherry picked”
Line 70: I am not sure if “incorrect” (per se the parameter namely occupancy is correct) is the right term here. I would rather say that they used an inadequate parameter, namely occupancy, as surrogate for detection probability of tiger scats.
Line 119: What does CV mean? Coefficient of variation!? You should write the full name the first time it appears in the text.

Reviewer 2 ·

Basic reporting

It is not often that one gets to review a paper such as this with all the ingredients of a scientific pot-boiler –famous names in wildlife ecology, potential statistical skulduggery, arguably a creature with the greatest mystique in human history, and national honour over its status at stake. Qureshi at al. may perhaps not have been so rhetorical over their criticism of the Gopalaswamy et al. paper had the media, ever alert for juicy stories on failure of the system, had not jumped into the fray, or if one of the co-authors had not given an interview to a prominent newspaper questioning tiger numbers and their monitoring in India. The latter’s paper may then have passed off as merely one that contributed to a staid scientific debate on animal population estimation methods. As matters stand, however, the paper has generated a lot of interest, publicity and controversy, and it would be in the interest of tiger conservation to set the record straight.

My task as a reviewer is not to do a thorough dissection of the Gopalaswamy et al. paper itself (which has gone through the process of peer-review), especially the theoretical basis of their statistical models, but to examine the claims in Qureshi et al. paper on the biases in the former which could render their conclusions infructuous. In other words, is there a strong enough case for publishing the Qureshi et al. paper in PeerJ?

Experimental design

Not relevant

Validity of the findings

Qureshi et al. make six basic observations to refute the conclusions of the Gopalaswamy et al. paper. I shall provide my assessment of whether these observations are valid or not for each of the six points.

a.Use of incorrect ecological parameters
Yes, it certainly seems that Gopalaswamy et al. have incorrectly used detection probability from tiger occupancy studies, rather than a double-blind observer design, to parametrize their theoretical model as stated by Qureshi et al.

b. Cherry picked references
Ostensibly, Gopalaswamy et al. have confined their model analyses to data from two sets of studies – Karanth and team, and Jhala and team – because these constitute the major tiger population estimation exercises from India in the past decade or more. However, Qureshi et al. have a valid point in that results of detection probability and coefficient of variation from at least two peer-reviewed papers on tiger estimation published in reputed journals have been (conveniently?) omitted by Gopalaswamy et al. thus rendering their conclusions less than robust.

c. Inappropriate and wrong data
The most telling criticism of the Gopalaswamy et al. paper is that they use invalid data (in Figure 5) on tiger count and tiger sign intensity because at half the sites their data have been collected years apart or wrongly reported (from one site). In particular, a 7-year time separation of tiger count and sign data from Bandipur would certainly make this data point invalid for deriving R2 as tiger abundance at this site could have varied considerably during this period. I do not have access to the Karanth & Kumar (2005) report from which a scat encounter rate of 1.99/10 km has been taken and reported as 3.6/10 km in Gopalaswamy et al. (2015). If this were true (and I am assuming that Qureshi et al. have been diligent in their homework and assertion), Figure 5 is indeed completely useless in deriving R2 for the tiger count-sign relationship, even if we condone the 2- or 3-year gaps in measurements at the other two sites, Pench and Melghat. Indeed, Tadoba would shift much further left along the X-axis and strengthen the value of R2 substantially. It is of course possible that Gopalaswamy et al. discovered an anomaly in the original figure for Tadoba reported in Karanth and Kumar (2005), but in this case they should have made this explicit in their paper. However, Qureshi et al. should also not take a R2 value of 0.642 (lines 180-181) for only four valid data points too seriously. They have already made their point about the inadequacy of Figure 5 in Gopalaswamy et al. In fact, it would be interesting if they could compute R2 after merely changing the “outlier” Tadoba value to 1.99.

d. Variability in tiger capture …. mark-recapture
While I cannot make a judgment about the reasons for low precision in Karanth et al. (2004), Qureshi et al. are correct in saying that Gopalaswamy et al. should not have restricted their model to examining only data from Karanth et al. (2004).

e. Repeating non peer-reviewed literature
Yes, the statement in Gopalaswamy et al. about the improbability of a 49% increase in tiger numbers over four years has been made without a proper assessment of what the 2006 and 2010 estimates actually represent. More sites were added for the tiger population estimation in 2010 and this should have been factored in while making statements questioning the rate of increase.

f. Propaganda that is not consistent with facts
It is unfortunate that these days even scientific journals need catchy “blogs” to capture the attention of their readers. In this case, the blog in Methods in Ecology and Evolution casting doubts on tiger population trends was unjustified. Subsequent articles in the media and even journal papers on the subject have slated towards a negative view of India’s tiger assessment. Even though this is not a scientific issue, Qureshi et al. are aggrieved by the negative publicity to the outcome of a rather challenging task of estimating tiger populations on such a large scale.

Additional comments

One piece of advice for Qureshi et al. is that they should not fall into the same trap of high rhetoric. They have made their case against Gopalaswamy et al. very clearly. I suggest some minor edits to remove the rhetoric – the edited paper would still retain its scientific punch!

Suggested list of edits:
Line 18: “double sampling”
Line 19: large scale animal surveys
Line 20: replace “elegantly” with “potentially” [after all, the statistical debate is not necessarily over]
Line 21: should be “Gopalaswamy”
Line 28: Could I suggest rewriting as “selectively-picked estimates from the literature”?
Line 29: suggest replacing “suspect” with “questionable”
Line 31: suggest “statistical design of large scale animal surveys”
Line 46: suggest “basic competence of scientists”
Line 48: “a peer reviewed publication”
Line 55: suggest “of the scientific method probably more than others”
Line 59: suggest “supposedly demonstrate”
Line 64: suggest “In particular, they claim the relationship…”
Line 67: suggest “selectively picked references from the literature” [the media can report this as cherry picked!]
Lines 88-90: Should be a single sentence with a comma (occupied site, while)
Lines 96-99: Should be a single sentence with a comma (Nichols et al. 2000), where two observers)
Line 107: “the use of occupancy detection”
Line 114: again “Selectively picked references”
Line 120: Should this be Karanth 2011a?
Line 139: Gopalaswamy
Lines 179-181: Suggest rewriting this part taking into consideration my suggestions given above (C. Inappropriate and wrong data)
Line 189: Which model is being referred to? Model 1 or Model 2?
Line 200: suggest “adoption of sound and practical methods”
Line 235: The reference states Jhala et al. 2014.
Line 238: same as previous
Line 243: I am wondering the word “propaganda” can be replaced with another, equally effective word? Distortion? Distorted reports? Misleading reports?
Line 243: suggest “MEE paper by Gopalaswamy et al.”
Line 248: Gopalaswamy
Line 252: I wonder if this line is needed? PeerJ editors can decide.

Reviewer 3 ·

Basic reporting

The language used in the manuscript is too emotive. I think it needs a substantial rewrite to focus on the issues of the science in Gopalaswamy et al. exclude suggestions of deliberate falsification of data - just say the data quality undermines their results.

Experimental design

I think a table of RAI (though not a full scale review) showing examples of fits between indexes and independent abundance estimates would greatly strengthen the author's case here and wouldn't involve a lot of extra work.

Validity of the findings

This manuscript focuses on responding to G et al's critique of their earlier work. It would be improved and strengthened by first concisely dealing with references to their work but then by comparing other RAI-based findings to broadening the perspective (see above).

Additional comments

Review: Twisted tale of the tiger: the case of inappropriate data and deficient science.
Quershi et al.

Summary:
The use of indexes in animal abundance surveys clearly divides researchers – in fact, I notice a number of the authors publishing in this literature, have co-authored both for and against views on the subject! The present manuscript is a (somewhat delayed) response to Gopalaswamy et al (2015) who - a) present a simulation model to support the assertion that index-based approaches are unreliable and ineffective, and b) use the simulation to argue that the results of Jhala et al (2011) with a reported high r2 between an index of tiger sign and tiger density - is likely to be spurious. Gopalaswamy et al critique was very personal and harsh in their criticism of Jhala et al., and given this I can understand why the present manuscript uses highly charged language. But I still I think the sections on scientific integrity and short-comings of peer review do not help the authors make an effective case. That said, I think a shortened and more focus response to G et al, is important and would help to clarify differences of view in the literature and perhaps could allay wider concerns about Indian tiger population estimates.


Main Points:

1) The manuscript would be greatly improved if the authors removed sections on scientific integrity etc. e.g. remove first two sentences in the abstract. Lines 41-46; lines 53-55; simply refer to manuscript published in MEE, remove prestigious journal etc. Remove all phrases like “cherry picking” “highly suspect” it needs to be written as If the authors of G et al. may have been in error but not scheming to undermine J et al.
2) A large part of G et al’s paper focuses on criticisms of RAIs. Could the authors here provide a table of examples of RAIs to give readers a better sense of typical fits to independent estimates of density. The most appropriate one I can think of which matches the case of Jhala et al 2011 is Funston et al 2010 (referred to in G et al, but not in the text strangely) – which uses a spoor count based RAI. But others use for photographic rates to provide RAIs e.g. O’Brien et al (2003); Rovero et al (2009) and Palmer et al (2018). Provide more evidence for better fits between indexes and actual density estimates. Demonstrating that indexes work in many settings, might be used to support the present manuscripts assertion that G et al’s model is based on overly pessimistic parameter values.
3) It was really good to see the authors clarify estimates of percentage of increase in the tiger population correcting for increased survey area. However, it would be also useful to know what percentage of the total tiger population estimate is obtained from the double-counting GIS extrapolated estimate. E.g. what perc is dervided from tigers in core areas where SECR methods were used compared to the remaining population. Reading the note published by Karanth et al (2011) one could assume that all (or most) the tiger estimates were based on extrapolations from indexes. Are we talking about <20%?
4) The authors make an interesting point about the use of occupancy statistics for G et al simulation. Could they clarify a bit further why this might not be appropriate?
5) It may also be worth pointing out that the Karanth et al (2004) paper comparing tiger density against prey density, was based on tiger density using MMDM method and not SECR. Might this have contributed to the poorer fit between the two?

·

Basic reporting

Detailed review provided in attachment.

Experimental design

Detailed review provided in attachment.

Validity of the findings

Detailed review provided in attachment.

Additional comments

Detailed review provided in attachment.

---

## Round 0.2 · Minor Revisions

We thank you for their patience in this process.

The revised version has been seen again by two of the original referees, and both agree that it is acceptable pending some final minor revisions. The response from Arjun Gopalaswamy remains scathing, which is not surprising. I do not think this precludes publication, but what I would ask is that you provide a final round of revision that at least attempts, in as much as is possible, to placate the major concerns of the author of the original paper.

Obviously there will remain points of flat out disagreement that cannot be resolved, beyond agreeing to disagree. However, where at least some common ground can be found, I would strongly encourage you to work further on those to reduce the region of friction between your viewpoints on the tiger population estimates, past and future.

Reviewer 2 ·

Basic reporting

no comments

Experimental design

adiquate

Validity of the findings

they are valid

Additional comments

The authors have addressed my comments satisfactorily.

Reviewer 3 ·

Basic reporting

Overall this revision is much improved and much clearer.

Experimental design

NA

Validity of the findings

See comments below

Additional comments

I find the revised version much improved, clearer and more concise and a more effective response to the original paper published by Gopalaswamy et al in MEE.
One point I would make is it would be great to have a paragraph from the authors, stating in broad simple terms, how they think the errors highlighted in this critique, have affected Gopalaswamy et al's results. I assume it means they are over critical of double sampling method, but it would be good to hear what the authors think in plain simple terms.. near the end of the paper.
Other than that I only have a number of minor edits to recommend (below)
L25 - abstract - show lags of (up to) several years?
L28 - change "wrong" with poor?
L76 - "tiger occupancy studies to surrogate detection probability" change to as a surrogate for..
L133 - "noticed several irregularities which invalidate these data as a scientific experiment." change to - which invalidates the use of these data to test this relationship?
L148-149 - Should you use Gopalaswamy et al 2012 in your revised figure or do you already?
L156 -" form" change to from
L174 - "experiment (IC-Karanth) to say the least are questionable" - I would say this more strongly - mentioning that this invalidates their conclusions.
L179 entire section: This section on variances in Karanth's tiger population estimates is the weakest bit to me. It seems like you are making criticising their work for the sake of it. Putting aside existing tensions, I wonder if this really contributes to this rebuttal. I think it doesn't unless you can make a clear case for why this contributes to the flaws in interpretation of G et al.
L220-1 - "for none of the national tiger status assessments (Jhala et al 2008, 2010, 2015) was tiger sign index solely used for modeling tiger abundance" rephrase - they should know that NTSAs are never based on tiger sign indexes alone..
L236-244- It is great to see that you now start to spell out the issue of what percentage of the population is based on actual individual tigers photographed, and then how many are extrapolated through SECR, It would STILL be good to know how many are estimated from the marginal areas through your double sample method that Gopalaswamy and co are so unhappy with. What is this percentage? This would great help readers and conservationists alike understand the scope of the the uncertainty.
L248-51- "It is surprising that with all of the above mentioned inappropriateness of data that do not meet the experimental setup for testing double sampling, incorrect theoretical concepts, and selective referencing of literature the Gopalaswamy et al. (2015a) paper passed through the peer review process." It is not that surprising - most reviewers/editors normally assume the data is carefully collected and compiled. I would cut this sentence, but go on with your discussion about your view that G et al has not made a useful contribution to the debate over the use of double sampling in very large-scale surveys
L263 - "(2015a,b) due to inappropriate scientific process and data, has not contributed on" - perhaps rewrite to say, it has that G et al's results have been misleading and therefore do not make a useful contribution to the wider debate ..

·

Basic reporting

No background or context is provided. I have suggested that the background information provided in the cover letter is moved to the main text. I have also provided ideas for the background in my previous review.

There is no hypothesis for the paper either. I have recommended that the authors ask a clear question right in the beginning.

Experimental design

This is not a primary research article. It is only a critique of an article by Gopalaswamy et al. (2015a,b). There really is no research question either.
There are many flaws in the arguments posed and several new evidences about the topic are not included (eg: Jhala et al. 2015). So there are many scientific contradictions.
There are many technical mistakes in the manuscript as well.

Validity of the findings

There are no data used in this manuscript. It is an attempted critique of Gopalaswamy et al. (2015a,b), with no real question behind that critique. The results are highly contradictory.

Additional comments

I would recommend that the authors pose their study question very carefully and clearly. And then I would recommend that they use sound statistical methods to demonstrate their case. Even if this is a critique of Gopalaswamy et al. (2015a,b) and not an original paper, they can write it as an original paper by demonstrating the whether the perceived data errors made in Gopalaswamy et al. (2015a,b) proved to be costly for the tiger sign and density relationship or not. For example, after Jhala et al. (2015) was published, which completely contradicted Jhala et al. (2011), they can show the similarity in the results between Jhala et al. (2015) and Gopalaswamy et al. (2015a,b). But they need to put in a lot of work and must not be fixed with a view that they have to verbally defend Jhala et al. (2011) and Jhala et al. (2015) even though they scientifically are at odds to one another.
* * *
INTRODUCTION

This is a review of the re-submitted version of the non-peer-reviewed manuscript titled ‘Twisted tale of the tiger: the case of inappropriate data and deficient science’ (Qureshi et al. 2018). I had earlier provided a detailed review of Qureshi et al. (2018) and it is also made available online (Gopalaswamy 2019). To avoid repeated use of pronouns and determiners, I will henceforth refer to this revised manuscript as Qureshi et al. (2018-V1). The corresponding author (Dr. Y. V. Jhala) has on behalf of the rest of the authors also written a covering letter to the PeerJ AE, with some relevant material on the topic and I will henceforth refer to this document as Jhala-Letter (2019).

Qureshi et al. (2018-V1) do not address any of the scientific contradictions and confusions discussed in Gopalaswamy (2019). The authors do not even provide scientific explanations about why they chose to ignore or accept each point raised by Gopalaswamy (2019). Consequently, all these scientific shortcomings persist in Qureshi et al. (2018-V1) and sure they do!

Qureshi et al. (2018-V1), however, have reduced the polemics that was present in Qureshi et al. (2018) as per recommendations from all the reviewers and the AE. But now, the authors have introduced a lot of new material related to the general topic about the failure of peer-review processes in Jhala-Letter (2019) instead of including all this material in Qureshi et al. (2018-V1). I have provided comments to the Jhala-Letter(2019) separately as an annotated document.


GENERAL COMMENTS

Since Qureshi et al. (2018-V1) ignore all the scientific recommendations and concerns expressed in Gopalaswamy (2019), virtually all the scientific inadequacies persist so my suggestion is for the authors to address/respond to them first.

The concerns raised and the excitement generated by Qureshi et al. (2018-V1), especially about the relationship between tiger sign indices and their respective densities, would have been scientifically relevant, for a brief period, about four years ago (i.e. between the time Gopalaswamy et al. (2015a,b) was published and the time Jhala et al. (2015) national tiger assessment report was released). Since Jhala et al. (2015) convincingly validates the findings of Gopalaswamy et al. (2015a,b), this part of the arguments developed in Qureshi et al. (2018-V1) becomes scientifically irrelevant.
To see this, I would advice the authors of Qureshi et al. (2018-V1) to study the mathematical derivations of Gopalaswamy et al. (2015a,b) carefully and then confront these derivations with the findings from Jhala et al. (2015) to see what they get before proceeding with their arguments and generating hasty conclusions.

The authors share some details via Jhala-Letter(2019) on how the journal Methods in Ecology and Evolution (a journal of the British Ecological Society) lacked the desired amount of transparency on the matter related to the publication of Gopalaswamy et al. (2015a,b). These are, of course, serious concerns about the procedures followed by such reputed journals and must be included as part of the main manuscript (Qureshi et al. 2018-V1) rather than be exclusively a letter to PeerJ. However, if they decide to include contents of Jhala-Letter(2019), they should share all the relevant facts and not a subset of them so that the readers can appreciate the flaws in the peer reviewed system, if any. Similarly, I would recommend that they include how the journal Journal of Applied Ecology, failed to detect data issues related to the publication of Jhala et al. (2011) – details mentioned in Gopalaswamy (2019), which has now proven to be costly (Jhala et al. 2015, Gopalaswamy et al. (2015a,b)).

More generally, as mentioned in Gopalaswamy (2019), I would emphasize the need for the authors to critically assess what their question really is. Without a clear scientific question, the authors will lose track of what they are seeking to really answer. And at this point, Qureshi et al. (2018-V1) reads more as a set of disjointed and random allegations rather than a set of coherent arguments seeking a specific answer to a scientific question.

SPECIFIC COMMENTS

I only have a few of technical additions to make to Gopalaswamy (2019):

l.62-74: This explanation for double-sampling is inadequate to the cause taken up by Qureshi et al. (2018-V1) there is also a population level variance associated with the index for the tiger cases. It is not limited to the ‘bias’ they are discussing here. So this section is completely irrelevant and misleading.

l.75-92: This refers to the identity described in Royle and Nichols (2003) for the relationship between p and r. The authors provide a flawed description of this identity. So I would suggest they first read up this paper fully and then work through the mathematics before putting anything in writing and causing a lot of confusion. If the authors do not want to put this effort, I would suggest they simply delete this entire section.

l. 101-104: It will be faulty to include any double-sampling citation here (whether it is Cochran 1937, Eberhardt and Simmons 1987, Pollock et al. 2002 or even Conroy and Carroll 2009). This is because all of these statistical approaches either involve a design-based approach (eg: Cochran 1937 or Eberhardt and Simmons 1987) or a model-based thinking (eg: Pollock et al. 2002 by finding a way to estimate detection probability). Since, India’s tiger assessments (Jhala et al. (2008), Jhala et al. (2011b) or Jhala et al. (2015)) are neither formally a design- or model-based in such a context, all these citations are irrelevant to the discussion here. And Jhala et al. (2011a) is purely an index-calibration experiment.

l. 239-242: These estimates are flawed (see the statistical sections in Gopalaswamy et al. 2015a,b) and juxtapose these on Jhala et al. (2015) to see what happens.

In general, if the authors are not comfortable with all the statistics involved in these studies, I would recommend they simply delete these sections and focus only on what they know well.

---

## Round 0.3 · accepted · Accept

Thank you for the latest round of revisions. With these changes, I am happy that the final paper stands as a robust and useful piece of critical work. I thank you for your patience and persistence during this process.